# Effect of Multiple Rounds of Enrichment on Metabolite Accumulation and Microbiota Composition of Pit Mud for *Baijiu* Fermentation

**DOI:** 10.3390/foods12081594

**Published:** 2023-04-09

**Authors:** Dong Li, Guangbin Ye, Xuyan Zong, Wei Zou

**Affiliations:** 1School of Life Science, Sichuan University of Science & Engineering, Yibin 644000, China; 320086002206@stu.suse.edu.cn (D.L.);; 2Liquor-Making Biotechnology and Application of Key Laboratory of Sichuan Province, Sichuan University of Science & Engineering, Yibin 644000, China

**Keywords:** Chinese strong-flavor *Baijiu*, pit mud, enrichment culture, volatile-acid production, high-throughput sequencing

## Abstract

Pit mud (PM) is the main component of *Baijiu* (traditional Chinese liquor), and its microorganisms are the primary sources of the aroma of Chinese strong-flavor *Baijiu* (SFB). Enrichment plays an important role in the selection of functional microorganisms in PM. Herein, the PM of SFB was submitted to six rounds of enrichment using clostridial growth medium (CGM), and changes in the metabolite accumulation and microbiota composition were evaluated. Based on the metabolite production and microbiota composition, the enrichment rounds were classified as the acclimation stage (round 2), main fermentation stage (rounds 3 and 4), and late fermentation stage (rounds 5 and 6). Species within the genus *Clostridium* dominated in the acclimation stage (65.84–74.51%). In the main fermentation stage, the dominant microbial groups were producers of butyric acid, acetic acid, and caproic acid, which included *Clostridium* (45.99–74.80%), *Caproicibacter* (1.45–17.02%), and potential new species within the order of *Oscillataceae* (14.26–29.10%). In the late stage of enrichment, *Pediococcus* dominated (45.96–79.44%). Thus, the main fermentation stage can be considered optimal for the isolation of acid-producing bacteria from PM. The findings discussed herein support the development and application of functional bacteria by bioaugmentation, and contribute to improving the quality of PM and SFB production.

## 1. Introduction

*Baijiu* is a traditional Chinese liquor, which can be classified into four basic flavor types based on the flavor compounds and brewing technology: strong-flavor, fen-flavor, Maotai-flavor, and rice-flavor. Among these, strong-flavor *Baijiu* (SFB) is the most commonly consumed across the largest geographical area, as well as constituting the largest market share in China. The brewing container in which SFB is produced as a cuboid cellar in which PM enclosed. During fermentation, *Zaopei* (fermented grains) is enveloped by PM, which is derived from long-term and continuous previously fermented PM [1]. Moreover, PM plays an important role in the formation of the flavor compounds of SFB [2,3].

The characteristic flavor substances in SFB are related to four major acids (i.e., acetic acid, butyric acid, lactic acid, and caproic acid) and four major esters (i.e., ethyl acetate, ethyl lactate, ethyl butyrate, and ethyl caproate). Ethyl caproate is the main component among the four major esters; its proportion accounts for 35–45% of the total esters [4]. The national standard GB/T 10781.1-2006 establishes the content requirements for ethyl caproate—the content of ethyl caproate in low-ethanol SFB (with ethanol content within 25–40%) should be within the range of 0.70–2.20 g/L, whereas in high-ethanol SFB, whose ethanol content can be within the ranges of 41–60% and 61–68%, it should be 1.20–2.80 g/L and 1.20–3.50 g/L, respectively [5]. Imbalances in the contents of these products can be attributed to imbalances in the structure of the PM microbial community. However, considering that the microorganisms associated with *Baijiu* might be derived from PM, *Zaopei* and *Daqu*, it is challenging to study the microbiota within this complex ecosystem. Therefore, the elucidation of fermentation imbalances is still a major issue in *Baijiu* research.

It is known that PM microorganisms play a pivotal role in the formation of SFB styles. In particular, SFB enterprises emphasize the importance of PM to the SFB brewing system, especially the importance of mature PM and PM in old pits [6,7,8]. Previous studies found that long-term acidic starch nutrition (with the pH in *Zaopei* between 2 and 5) and selective pressure for strict anaerobic or facultative anaerobic fermentation encourage the directional accumulation of microorganisms with specific aroma-producing functions in PM [9,10,11].

High-throughput sequencing technology can provide key information on the succession patterns of PM microorganisms, which might enable the confirmation of functional microorganisms. Tao et al. conducted comparative studies on the prokaryotic biodiversity of PMs with different ages by using pyrosequencing technology, which revealed that the prokaryote community reached a relatively mature stage as the PM age increased. *Clostridium* and *Ruminococcaceae* were found in higher abundance and have the ability to metabolize caproic acid into ethyl caproate [12]. Hu et al. compared the microbial composition of PMs of different quality using the Illumina MiSeq sequencing platform, and their results revealed that 11 bacteria general (i.e., *Lactobacillus*, *Ruminococcus*, *Caloramator*, *Clostridium*, *Sedimentibacter*, *Syntrophomonas*, *Sporanaerobacter*, *Pelotomaculum*, *T78*, *Prevotella*, and *Blvii28* group) and 6 archaea genera (i.e., *Methanobacterium*, *Methanobrevibacter*, *Methanosaeta*, *Methanoculleus*, *Methanosarcina*, and *Nitrososphaera*) were among the core microorganisms in the PM [13]. In addition, high-throughput sequencing results revealed that a large number of functional microorganisms in PM are difficult to cultivate. The composition and ecological function of microorganisms in PM can be analyzed, to a certain extent, by comparing the high-throughput sequences obtained from PM samples and by collecting ecological-function information of similar microorganisms. However, due to the complexity of the mixed fermentation system in PM, research on the functional microorganisms in PM still relies on the study of cultivable microorganisms. So far, eight novel species have been isolated from PM [14,15,16,17,18,19]. In addition, it was reported that the main acid-producing functional microbial groups in PM are *Clostridium* [20,21,22,23,24,25], *Oscillibacter* [26], *Caproicibacterium*, and *Caproiciproducens* [27,28]. It is estimated that the proportion of currently known culturable microorganisms isolated from PM is 3–33% of the total number of microbial taxa presumably harbored in PM [13,29].

Enrichment culture is an effective method for the isolation of difficult-to-culture microorganisms from environmental samples. Wu et al. isolated several bacteria with the ability to degrade environmental pollutants (including lindane and endosulfan) from their enrichment culture [30]. Moreover, Pernicova et al. added activated sludge to the enrichment medium (MSM), which was rapidly detected by ATR-FTIR, and *Bacillus* with polyhydroxyalkanoate (PHA)-producing ability was also isolated [31]. In addition, it was shown that species within the genus *Clostridium* were the dominant microorganisms in the PM. Moreover, *Clostridium* species have a strong ability to produce short-chain fatty acids (SCFAs), such as butyric acid, acetic acid, and caproic acid, as well as the ability to produce spores and thermoresistance [32]. Therefore, applying heat-shock pretreatment to PM enables selectivity for enriching certain microbial groups, such as *Clostridium* and *Bacillus* [33]. Clostridial growth medium (CGM) is commonly used for isolating clostridia and studying anaerobic fermentation characteristics [34]. Yuan et al. enriched *Clostridium butyricum* YD-4 from PM using CGM, and revealed that the butyric-acid yield was 3.44 g/L [25]. In addition, Qian et al. isolated 54 microbial strains from PM using CGM [35]. In other studies, the mixed bacteria system in PM was enriched using glucose-based CGM and heat treatment, which resulted in a high yield of caproic acid (6.68 g/L) and butyric acid (2.46 g/L) [36].

In the present study, PM was enriched using a clostridial growth medium and submitted to heat shock. The fermentation products and microbial-community composition of enriched fermented broth (EFB) in different rounds of the enrichment were studied to determine the effects of the enrichment medium, enrichment rounds, and heat-shock treatment on the metabolites and microbial succession patterns in the PM. The obtained findings are of great significance for the development of the functional fermentation broth of PM and the isolation of difficult-to-culture microorganisms.

## 2. Materials and Methods

### 2.1. Sample Collection and Treatment

The PM samples were collected at a famous distillery in Mianyang City, Sichuan province, China. They were transported to the laboratory on ice and stored at 4 °C until further analysis. In the first round of enrichment, 50 g of PM was divided into two flasks containing 250 mL of CGM, each containing 25 g of PM; prepared mixtures were labeled as P13-1 and P14-1, which respectively referred to samples not submitted to heat-shock treatment after inoculation and samples treated at 80 ◦C in a water bath for 15 min. Next, prepared samples were placed in an incubator at 35 °C for 7 days, and continuous subculture was carried out with 10% of the initial inoculum.

A total of six enrichment rounds were conducted in this experiment, and were labeled as P13-1, P13-2, P13-3, P13-4, P13-5, P13-6 and P14-1, P14-2, P14-3, P14-4, P14-5, P14-6, respectively. It should be noted that only P14-1 was subjected to heat-shock treatment; the subsequent subcultures were not heat-treated. Subsequently, samples of EFB from enrichment rounds 1–6 were stored in a refrigerator at −20 °C for subsequent experiments.

The composition of CGM used herein was (per liter): peptone, 10 g; yeast powder, 3 g; glucose, 5 g; soluble starch, 1 g; CH_3_COONa, 3 g; and cysteine hydrochloride, 0.5 g; oxygen-free distilled water was added to obtain 1000 mL at natural pH [25,37]. The prepared CGM was deoxygenated by nitrogen blowing [38] and then autoclaved at 121 °C for 20 min.

### 2.2. Flavor Components of PM-EFB

Headspace solid-phase micro-extraction (HS-SPME) was performed to extract trace volatiles from EFB samples. Briefly, 20 mL of fermentation broth in a centrifuge tube was mixed with H_2_SO_4_ 2 mol/L (adjust pH to 2), and absolute ethanol was added and thoroughly mixed to obtain 25 mL of final mixture, which was subsequently centrifuged at 12,000 r/min for 5 min at 4 °C. Next, the supernatant was obtained and filtered through a 0.2-μm microporous membrane to obtain the test solution [39]. An aliquot of the test solution was injected in a headspace bottle, and 10 g of NaCl and 10 µL of the internal standard butyl acetate (84 mg/mL) were added. Next, the mixture was equilibrated at 55 °C for 15 min and extracted by headspace adsorption for 30 min. Subsequently, the headspace needle was inserted into the sample inlet of a gas chromatography–mass spectrometer (GS-MS) (7890A-5975B, Agilent Technology Co., Ltd., Santa Clara, CA, USA), the sample was injected at 230 °C, and volatile components were analyzed [16]. Volatile components were extracted using a 50/30-µm divinylbenzene/carboxen/polydimethylsiloxane (DVB/CAR/PDMS) fiber (ANPEL Laboratory Technologies Inc., Shanghai, China), which was heated at 250 °C for 1 h to remove impurities prior to use. The GC conditions were as follows: capillary column, DB-Wax (60 m × 250 µm × 0. 25 µm); carrier gas, He; flow rate, 1 mL/min; inlet and detector temperature, 230 °C. The GC-temperature program was as follows: initial temperature was 60 °C, and held for 1 min; it was then increased to 180 °C at a rate of 8 °C/min, and maintained for 2 min; and it was finally increased to 230 °C at a rate of 15 °C/min, and maintained for 5 min. The concentration of volatile compounds in EFB samples was calculated based on the percentage of the corresponding peak area [40].

The MS conditions were as follows: interface temperature, 230 °C; quadrupole temperature, 150 °C; ionization mode, electron-bombardment source (EI); full-scan detection mode; ionization energy, 70 eV; scanned mass range, 20–550 amu. Mass spectra obtained from EFB samples were compared with standard mass spectra provided by National Institute of Standards and Technology (NIST) 05a.L database (Agilent Technology Co., Ltd., USA), and mass spectra with a matching degree greater than 80 were selected for qualitative analysis [41]. The experiment was repeated at least three times independently.

### 2.3. DNA Isolation

Genomic DNA was isolated from EFB samples using an extraction method proposed elsewhere [42]. Specifically, EFB was collected in a 50-mL centrifuge tube, and submitted to centrifugation (Lynx-6000, Thermo Fisher) at 12,000× *g* for 1 min. The obtained supernatant was decanted, and bacterial cells were collected after 2–3 rounds of centrifugation. Next, 13.5 mL of DNA-extraction buffer (100 mmol/L Tris-HCl, pH 8.0, 100 mmol/L EDTA, pH 8.0, 100 mmol/L NaCl, 1% CTAB, 2% SDS) and 200 µL lysozyme (20 mg/mL) were added to the sedimented cells in a 50-mL centrifuge tube, followed by incubation at 37 °C in a water bath for 30 min (mixtures were gently inverted at each 10-min interval). Next, 1.5 mL of 20% SDS preheated at 65 °C was added to the mixture, followed by homogenization and incubation at 65 °C in a water bath for 30 min. After centrifugation at 5000× *g* at 4 °C for 10 min, the obtained supernatant was transferred to a new 50-mL centrifuge tube, and an equal volume of phenol chloroform isoamyl alcohol (25:24:1) was added to the mixture and homogenized. After standing for 10 min, the mixture was centrifuged at 13,000× *g* at 4 °C for 20 min, and the upper water phase was transferred to a new 50-mL centrifuge tube and extracted thrice. Subsequently, 0.6× volume of isopropanol was added to the mixture, and the precipitate was obtained at room temperature for 10 min. After centrifugation at room temperature at 13,000× *g* for 20 min, the supernatant was discarded, and the residual isopropanol was removed. The precipitate was transferred to a 1.5-mL centrifuge tube, 1 mL of freshly prepared 70% ethanol was added to the tube, and the mixture was left to stand for 10 min. Next, the mixture was submitted to centrifugation at 4 °C at 13,000× *g* for 15 min (Eppendorf^®^ Centrifuge 5430, Eppendorf AG, Hamburg, Germany). Once the last centrifugation step was repeated, tubes were left open to allow residual ethanol to air dry. Finally, the obtained DNA pellet was rehydrated in 100 µL of sterile water and stored at −20 °C. After evaluating the quality of extracted genomic DNA samples by agarose-gel electrophoresis, *16S rRNA* gene-amplicon sequencing was performed.

### 2.4. 16S rRNA Gene-Amplicon Sequencing

Library construction and amplicon sequencing were conducted by Shanghai Majorbio Biotechnology Co., Ltd. (Shanghai, China). The hypervariable V3–V4 region of the bacterial *16S rRNA* gene was targeted for amplifications using the primers 338F (5′-ACTCCTACGGGAGGCAGCAG-3′) and 806R (5′-GGACTACHVGGGTWTCTAAT-3′) [43] on an Illumina MiSeq platform. Raw fastq files were demultiplexed and quality-filtered using Trimmomatic software [44]. Demultiplexed paired-end sequences were denoised, merged, and clustered into amplicon-sequence variant (ASV) using dada2/deblur program in QIIME2 “https://qiime2.org/ (accessed on 26 February 2020)” [45]. Taxonomic classification of ASVs was performed using QIIME2 with a Naive Bayes classifier previously trained on the SILVA 138 database considering 99% sequence similarity “http://www.arb-silva.de (accessed on 26 February 2020)”. Microbial diversity analysis was performed using the cloud platform “https://cloud.majorbio.com/ (accessed on 11 March 2022)”. The bacterial flora heatmap (Spearman-correlation heatmap) and CCA were analyzed by the R language (vegan package).

The *16S rRNA* genome sequences of 12 samples were deposited in NCBI under the accession numbers SAMN32722911, SAMN32722912, SAMN32722913, SAMN32722914, SAMN32722915, SAMN32722916, SAMN32722917, SAMN32722918, SAMN32722919, SAMN32722920, SAMN32722921, and SAMN32722922, respectively.

## 3. Results

### 3.1. Volatile Compounds in PM-EFB

The volatile components in the EFB during six enrichment rounds (with 12 samples in total) were identified by HS-SPME-GC-MS. In particular, changes in the contents of the organic acids in the EFB in different rounds of enrichment were evaluated (Figure 1A). The total acid, acetic acid, butyric acid, and caproic acid contents were within the ranges of 18.22–70.10 g/L, 9.41–29.21 g/L, 7.78–33.37 g/L, and 5.61–17.47 g/L, respectively. During the six rounds of enrichment, the organic acid contents in the EFB initially increased and then decreased. The total acid, acetic acid, and butyric acid contents peaked between rounds 3 and 4 of the enrichment, whereas the caproic acid content peaked in round 2. More specifically, during rounds 1–4 of the enrichment, the total acid contents in the P13 increased from 36.17 g/L to 70.10 g/L, whereas those of the acetic acid and butyric acid increased from 9.41 g/L to 29.21 g/L and 19.69 g/L to 33.37 g/L, respectively. Until round 6 of the enrichment, the total acids, acetic acid, and butyric acid decreased to 34.96 g/L, 11.11 g/L, and 16.79 g/L, respectively. In the P13, the caproic acid content initially increased from 7.57 g/L to 17.47 g/L, during rounds 1–2 of the enrichment, and then decreased to 6.48 g/L at the end of round 6 of the enrichment. In the P14, the total acid, acetic acid, and caproic acid contents in the P14 increased from 18.22 g/L to 61.39 g/L, 9.57 g/L to 26.12 g/L, and 5.61 g/L to 9.69 g/L, respectively, during rounds 1–3 of the enrichment, and then decreased to 30.58 g/L, 10.63 g/L, and 6.10 g/L, respectively, at the end of round 6 of the enrichment. Simultaneously, the butyric acid content in the P14 increased from 7.79 g/L to 26.19 g/L during rounds 1–4 of the enrichment, and then decreased to 14.11 g/L at the end of round 6 of the enrichment.

In the first enrichment round, the total acids accumulated in the P13-1 and P14-1 reached significantly different concentrations (*p* ≤ 0.01) due to the influence of the PM and the heat shock. In the subsequent enrichment rounds, however, although the total acid content in the P13 was slightly higher than that in the P14, the difference was not significant (Figure 1B). Thus, it can be speculated that the heat-shock treatment had little influence on the acid production during the PM-enrichment rounds. In contrast, differences in the total acid content between the different enrichment rounds were observed (Figure 1C). In particular, a significant difference in P13 and P14 was found between rounds 1 and 6 of the enrichment and rounds 2, 3, 4, and 5 (*p* ≤ 0.01). In addition, a significant difference was found between rounds 4 and 5 of the enrichment (*p* ≤ 0.01). Overall, it can be inferred that the enrichment rounds contributed more significantly to the enrichment of the organic acids in the PM compared to the heat-shock treatment.

### 3.2. Bacterial Community Diversity in PM-EFB

In total, 709,230 sequences were obtained from the sequencing of 12 EFB samples, with an average length of 382.16 bp for all the samples, and the samples contained 48,510–68,658 sequences. The sequencing coverage of each sample was greater than 99.9%, and the sparse curve was close to saturation, thus indicating that the sequencing depth of the amplicon library adequately reflected the composition of the bacterial communities in the evaluated samples (Appendix A). The Shannon diversity index was the highest (3.37) in the P13-1, which contained PM and was not submitted to heat-shock treatment. Conversely, the Shannon diversity index was decreased significantly (1.48) in the P14-1, which contained PM and was heat-treated, thus indicating that the heat shock inactivated a large number of thermo-sensitive microorganisms in the PM. Except for the sample P13-6, which showed a rapid increase, the overall trend in the Shannon indexes in the samples of the two treatments within rounds 2–6 of the enrichment was an initial increase followed by a decrease. The Shannon indexes of the round-3 samples were the highest, at 2.87 (P13-3) and 2.84 (P14-3), respectively. Moreover, the coefficient of variation of the Shannon indexes in the two samples from the same round was smaller than the difference between the rounds (Appendix A and Figure 2A).

Furthermore, a PCoA analysis based on the ASV data showed that the 12 samples grouped into four distinguishable clusters (Figure 2B), i.e., P13-2 and P14-2 clustered together; P13-1, P13-3, and P14-3 clustered together; P14-1, P13-4, and P14-4 clustered together; and P14-5, P14-6, P13-5, and P13-6 clustered together. Considering the influence of the PM and the heat shock on the round-1 samples, only around 2–6 samples were analyzed. It was found that the samples from the same enrichment round and submitted to different treatments tended to cluster together. Thus, it can be hypothesized that the heat-shock pretreatment affected the succession pattern of the microbial community to a lesser extent than the enrichment rounds. Moreover, based on the differences in acid production between the different enrichment rounds, the enrichment process was divided into three stages, i.e., the acclimation stage (round 2), the main fermentation stage (rounds 3–4), and the late fermentation stage (rounds 5–6).

### 3.3. Changes in Bacterial Community Composition

The sequences with more than 0.1% valid sequences in the 12 EFB samples were annotated. At the phylum level, the bacterial community in the EFB was mainly composed of *Firmicutes*, and *Bacteroidetes* (10.93%) was found only in sample P13-1 (Figure 3A). In addition, the sequence alignment of 58 ASVs obtained from the 12 samples and whose proportion was greater than 0.1% showed that 44 of the ASVs had a similarity ≥ 97% with the sequences of the culturable microorganisms available on the NCBI database, whereas 14 of the ASVs had a similarity < 97% with these sequences NCBI, thus indicating that there were potentially new species in the EFB. Furthermore, we merged the ASVs with a sequence similarity of ≥ 97% into the same OTU, and the fifty-eight ASVs were grouped into twenty-nine OTUs belonging to nine genera, which were assigned to three families, including twenty-one known species and eight unclassified species (Figure 3B and Appendix A). Twelve representative ASVs, ASV134, ASV95, ASV104, ASV135, ASV109, ASV103, ASV196, ASV 136, ASV152, ASV137, ASV101, and ASV96, were the dominant species, which accounted for 61.39–95.3% of the samples. These dominant ASVs were mainly composed of *Clostridium*, *Caproicibacterium*, *Pediococcus*, *Lacticaseibacillus*, *Weizmannia*, and potential novel species within the order Oscillatoriaceae (Figure 3B and Appendix A).

In addition, in order to better understand the composition and succession patterns of the dominant microbial communities in each enrichment round (abundance ≥ 1% in one of 1–6 rounds and more dominant than other rounds), we inversed the community-succession pattern in each sample at the species level based on the ASV-alignment data (Figure 3B). In the first round of the enrichment, samples P13-1 and P14-1 were greatly affected by the PM and heat-shock treatment, which resulted in significant differences in microbial composition. Comparing the differences in microbial composition between samples P13-1 and P14-1, it was possible to determine which microorganisms are sensitive to heat shock. Specifically, the dominant species in the P13-1 were *Clostridium neuense* (45.46%), *Clostridium guangxiense* (10.32%), *Clostridium diolis* (6.31%), *Lactobacillus acetotolerans* (5.45%), *Clostridium liquoris* (4.52%), *Clostridium algifaecis* (3.89%), *Clostridium fermenticellae* (3.33%), *Clostridium kogasense* (3.26%), *Clostridium luticellarii* (2.24%), *Clostridium tepidum* (1.08%), and potential new species found within the orders *Bacteroidales* and *Oscillospiraceae*, accounting for 10.93% and 1.27% of the abundance, respectively. In contrast, the dominant microorganisms in the P14-1 were *C. guangxiense* (58.00%), *Weizmannia coagulans* (34.62%), *C. fermenticellae* (3.10%), *C. luticellarii* (1.65%), and *C. kogasense* (1.08%). By comparing the differences in microbial community composition between samples P13-1 and P14-1, it was revealed that the heat-shock treatment inhibited several species within the genera *Clostridium* and *Lactobacillus*, as well as within the order *Bacteroidales*, which are probably thermolabile microorganisms, or organisms not adapted to CGM, except for *C. guangxiense*, *C. fermenticelles*, *C. luticellarii*, and *C. kogasense*, which occurred simultaneously in samples of the two treatments.

In the second round of the enrichment (the adaptation stage), 13 microbial groups were found simultaneously in the P13-2 and P14-2 (Figure 3B and Appendix A), and the dominant microorganisms in this enrichment round included *C. kogasense* (58.25% and 56.92%, respectively), *C. fermenticellae* (16.26% and 8.92%, respectively), *C. liquoris* (4.31% and 1.72%, respectively), and potential new species within the order *Oscillospiraceae* (1.07% and 1.39%, respectively). In addition, by comparing the community-succession pattern between the P13-1 and P13-2 with that between the P14-1 and P14-2, it was revealed that, independently of the application of heat-shock treatment, the microbial communities in these samples were dominated by *C. kogasense* and *C. fermenticelles* in round 2 of the enrichment, which showed that the CGM specifically contributed to the changes in the microbial community of the PM.

In rounds 3–4 of the enrichment (the main fermentation stage), 11 microbial groups were commonly found in the P13 and P14 (Figure 3B and Appendix A). The dominant microbial groups were *C. algifaecis* (1.01–22.28%), *Caproicibacterium amylolyticum* (1.03–15.97%), *C. guangxiense* (12.69% and 44.50%, respectively), *C. fermenticellae* (0.78–20.09%), *Lacticaseibacillus casei* (1.53–15.72%), *Clostridium pabulibutyricum* (0.20–25.73%), and potential new species within the order *Oscillospiraceae* (17.23–27.50%). In addition, it was observed that the dominant microorganism in the main fermentation stage was still *Clostridium*, followed by *Caproicibacterium*, *Lacticaseibacillus*, and potential new species. Compared with the microbial community composition in the adaptation stage (round 2 of the enrichment), the diversity in the acid-producing microorganisms in the main fermentation stage increased.

Finally, in rounds 5 and 6 of the enrichment, eight microbial groups were commonly found in the P13 and P14 (Figure 3B and Appendix A). The dominant microbial group included *Pediococcus acidilactici* (45.96–79.44%, respectively) and potential new species within the order *Oscillospiraceae* (6.61–29.05%). Thus, it can be seen that *Pediococcus* was the dominant bacteria in this stage, but the reason for this phenomenon could not be determined.

## 4. Discussion

### 4.1. Correlation between Dominant Acid-Producing Microorganisms and Organic Acid Contents

Ethyl caproate, ethyl acetate, and ethyl butyrate were the main volatile compounds in the SFB. The precursors of caproic acid, acetic acid, and butyric acid are mainly produced by acid-producing functional bacteria in PM and *Zaopei*. Simultaneously, caproic acid and butyric acid are mainly produced by PM microorganisms [46]. The caproic-acid-producing bacteria in PM previously reported in the literature include *Caproiciproducens galactitolivorans* [47], *C. amylolyticum* [14], *Ruminococcaceae bacterium* CPB6 [48], *Caproiciproducens fermentans* [49], *C. fermenticellae* [16], *Ruminococcaceae bacterium* BL-4, *Clostridium* sp. BL-3 [20], and *Clostridium kluyveri* DSM 555 [50]. These caproic acid-producing bacteria can utilize a variety of precursors to synthesize caproic acid through reverse beta-oxidation using ethanol or lactic acid as precursors in the form of acetyl-CoA [36,51]. In particular, *R. bacterium* CPB6 was shown to synthesize caproic acid from lactic acid [48]. It was demonstrated that *C. galactitolivorans* BS-1 produces n-caproic acid through chain elongation [22,52]. In addition, a large number of species within the genus *Clostridium*, often found in PM and fermented grains, can decompose starchy raw materials or metabolize glucose to produce mixed acids, such as butyric acid and acetic acid, under anaerobic conditions. Based on the findings of the present study, as well as the information on the newly reported species *Clostridium acidigenes*, it was shown that the EFB mainly contained *C. guangxiense* [53], *C. algifaecis* [54], *C. kogasense* [55], *Clostridium autoethanogenum* [56], *Clostridium chromiireducens* [57], *Clostridium butyricum*, *Clostridium diolis* [58], *Clostridium liquoris* [59], *Clostridium luticellarii* [60], and others. Moreover, it was demonstrated herein that the enrichment rounds led to an enrichment of *C. fermenticellae* (with ASV101 as a representative), *C. amylolyticum* (with ASV152 and ASV125 as representatives), *C. guangxiense* (with ASV95 as a representative), *C. algifaecis* (with ASV109 as a representative), and *C. pabulibutyricum* (with ASV136 as a representative). In addition, the enrichment stage in which the highest diversity of acid-producing bacteria and the highest total acid content were found was the main fermentation stage. Therefore, the main fermentation stage is the best for pure strain isolation, and it is also considered the best stage for harvesting the products of mixed bacterial fermentation.

Subsequently, we aimed to explore the relationship between the acid-producing functional microorganisms and the metabolites in the PM-enriched samples. To achieve this, we further obtained new combinations by grouping 29 OTUs with similar acid-producing functions based on previously reported data to perform a canonical correspondence analysis and construct a Spearman-correlation heat map between the composition of the functional microbial community and the metabolites identified in the samples (Figure 4). It was revealed that the EFB samples with positive correlations with the volatile acids were obtained during rounds 2 to 4 of the enrichment, while the samples obtained in the later enrichment stages showed negative correlations with the volatile acids (Figure 4A). Moreover, the Spearman correlation thermogram (Figure 4B) showed that the presence of microbial groups (C_a) with caproic-acid-producing ability in the EFB samples was positively correlated with the caproic acid contents in the samples. Among these microbial groups, *C. fermenticellae* (with ASV101 as a representative) and *C. amylolyticum* (with ASV152 and ASV125 as representatives) were important constituents of the microbial community for caproic acid accumulation. In addition, microorganisms (AB_a) such as *C. guangxiense* (mainly ASV95), *C. algifaecis* (mainly ASV109), *C. kogasense* (mainly ASV104), and *C. pabulibutyricum* (ASV136), which can produce acetic acid, butyric acid, or mixed acid, showed a positive correlation with the contents of acetic acid and butyric acid in the EFB samples.

Lactic acid is a non-volatile acid that cannot be detected by GC-MS. Although ethyl lactate synthesized from lactic acid is an important volatile compound in SFB, relevant studies suggested that the excessive production of lactic acid inhibits the formation of the main flavor (ethyl caproate) of SFB [61]. In this study, *L. casei*, *Levilobacillus brevis*, *P. acidilactici*, *W. coagulans*, and *Secundiobacillus mixtipabul* were enriched. These microbial groups (L_a) were negatively correlated with acetic acid, butyric acid, and caproic acid. Relevant studies also reported that, when found in high abundance, lactic acid bacteria can inhibit clostridia and caproic acid production by lowering environmental pH, which leads to the accumulation of lactic acid and/or the secretion of some bacteriocins [12].

In addition, it was found that the potentially dominant communities (ASV 142, ASV 103, ASV 116, ASV 119, ASV 153, ASV 145, and ASV 148) newly identified in the EFB samples were also positively correlated with the caproic acid content. Among these, the ASV 142, ASV 103, ASV 116, and ASV 119 shared 94.55–96.53% similarity with *C. amylolyticum*, whereas the ASV 153, ASV 145, and ASV 148 shared 93.87–94.85% similarity with *C. galactitolivorans*; therefore, it can be speculated that these may potentially constitute novel species with caproic-acid-production ability. In addition, the ASV135 and ASV190 were positively correlated with the acetic acid and butyric acid contents. The ASV135 and ASV190 shared 92.89–93.38% similarity with the *16S rRNA* gene sequence of *O. ruminantium* GH1. Related studies also showed that *O. ruminantium* GH1 can use D-glucose, D-ribose, and D-xylose to produce butyric acid [26]. Therefore, it can be speculated that ASV135 and ASV190 may have the ability to produce butyric acid. Interestingly, the potential novel species *Bacteroidetes* was found only in the P13-1 samples, which were negatively correlated with the contents of acetic acid, butyric acid, and caproic acid. Thus, these species may have other functions.

### 4.2. Analysis of Community-Succession Pattern and Enrichment Stage of Specific Acid-Producing Microorganisms

During our multi-round enrichment fermentation, the microbial community compositions in samples P13-1 and P14-1 significantly differed due to the application of heat-shock treatment in round 1 of the enrichment in samples P14-1 (Figure 3B). Relevant studies found that heat treatment can reduce the abundance of certain non-spore-forming microorganisms, such as *Lactobacillus*, a few clostridia, and some organisms within the genus *Bacteroidetes* (e.g., *Bacteroides*, *Odoribacter*, and *Alloprevotella*) [62,63]. Comparing the microbial community structures in rounds 2–6 of the enrichment, it was shown that the CGM had strong selectivity for the anaerobic microorganisms in the PM. Among these, the dominant microbiota capable of metabolizing volatile acids in samples P13-2 and P14-2 during the adaptation stage of the enrichment were *C. kogasense*, *C. liquoris*, and *C. fermenticellae*, as well as ASV116 and ASV119 (Figure 5A and Figure 4B). In the main fermentation stage, the abundance of *C. kogasense* started to decrease, while those of *C. fermenticellae*, *C. amylolyticum*, *C. guangxiense*, *C. algifaecis*, *C. pabulibutyricum*, and the potential new caproic-acid-producing species ASV142, ASV103, ASV135, ASV190, ASV 153, ASV145, and ASV148 were greatly increased, accompanied by acid-production and pH decreases (Figure 5B and Figure 4B). The replacement of most of the dominant clostridia in the main fermentation stage may have been related to the fact that *C. guangxiense* has strong acid tolerance compared to *C. kogasense* [57,59]. In the late stage of fermentation, the abundance of these dominant volatile-acid-producing microorganisms decreased sharply, and *Pediococcus* (from the family *Lactobacillaceae*) became the dominant microorganism (Figure 5C). In general, although the differences in microbial community composition in samples P13-1 and P14-1 were significant at the initial stage of the enrichment due to the effect of the heat-shock treatment, the dominant microbiota in both treatment groups were constituted by clostridia and potential new species (within the order *Oscillospiraceae*) during the adaptation stage based on the screening conducted using the CGM, which was also accompanied by an abundant accumulation of caproic acid. Based on the dominant microbiota in round 2 of the enrichment, five known bacterial species (from *Clostridium* and *Caproicibacterium*) and seven potential new species (within the order Oscillatoriaceae, accounting for 69.2–82.75% of the total in this stage) with caproic-acid-producing ability were found to be enriched by the programmed orientation during the main fermentation stage; the *P. acidilactici* was enriched in the late stage of the fermentation.

Furthermore, the succession pattern of these dominant acid-producing microbial groups was characterized by an initial increase followed by a decrease, which was similar to the microbial-community-succession pattern in *Zaopei* during the SFB fermentation. Using the known SFB-fermentation process, some studies found that the *Clostridium* in *Zaopei* increased initially, then decreased with the increase in fermentation time, and dominated mid-fermentation (7–30 days), whereas *Lactobacillus* mainly dominated in the late fermentation stage [46,64]. The succession pattern of the dominant *Clostridium* in this experiment was similar to the results of the studies cited above (Figure 5A,B). However, the difference is that the dominant bacteria enriched in the later stage of fermentation in this experiment is *Pediococcus*, which may be caused by the different starting points of *Baijiu* fermentation and enrichment liquid fermentation. (Figure 5C). Moreover, it was shown that, during SFB fermentation, the contents of volatile acids in *Zaopei* increase sharply mid-fermentation and then slowly increase in later stages [65,66,67]. A similar pattern was observed herein for the volatile-acid accumulation (Figure 5D). Therefore, the results of the present work can be a miniature summary of SFB fermentation.

Finally, the enrichment of PM has an important effect on the improvement of its quality. Good SFB is produced from old PMs, and the transition from new PM to mature, high-quality, old PM requires considerable time. Studies that compared PMs of different ages (5–100 years) found that the abundance of clostridia and bacteria from the family *Ruminococcaceae* in the PMs increased with age, which resulted in the greater production of volatile acids (especially acetic acid and caproic acid), thus improving the quality of *Baijiu* [68,69]. Therefore, the enrichment of clostridia in PM can provide a shortcut for the rapid maturation of new PMs.

## 5. Conclusions

In the present study, PM samples were submitted to heat-shock treatment and continuously subcultured in CGM in order to enable the exploration of the succession patterns of dominant microbial communities and metabolite production during enrichment. It was found that the heat-shock pretreatment affected the microbial diversity and abundance in samples P13-1 and P14-1, but no significant effect was observed in the subsequent enrichment rounds. Moreover, volatile acids (i.e., acetic acid, butyric acid, and caproic acid) were significantly enriched in the main fermentation stage of the enrichment. In addition, throughout the fermentation-enrichment process, *Clostridium* and *Caproiciproducens* with the ability to produce acetic acid, butyric acid, or caproic acid were found to be significantly enriched in the adaptation stage, as well as in the main fermentation stage. Interestingly, in the late stage of the fermentation enrichment, *Pediococcus* was found to be significantly enriched. Thus, it could be considered that the batch enrichment of PM was akin to a miniature SFB-brewing system, which can indirectly reflect the succession patterns of microorganisms and volatile-acid production in SFB during fermentation. Thus, the knowledge obtained on the dominant known bacterial groups and potential new species in EFB samples might support the development and functional application of SFB-strain resources. Finally, the present study provides a theoretical and practical basis for the enrichment and isolation of functional microorganisms, as well as the development and application of functional bacteria through bioaugmentation, thereby improving the quality of PM and SFB production.

## Figures and Tables

**Figure 1 foods-12-01594-f001:**
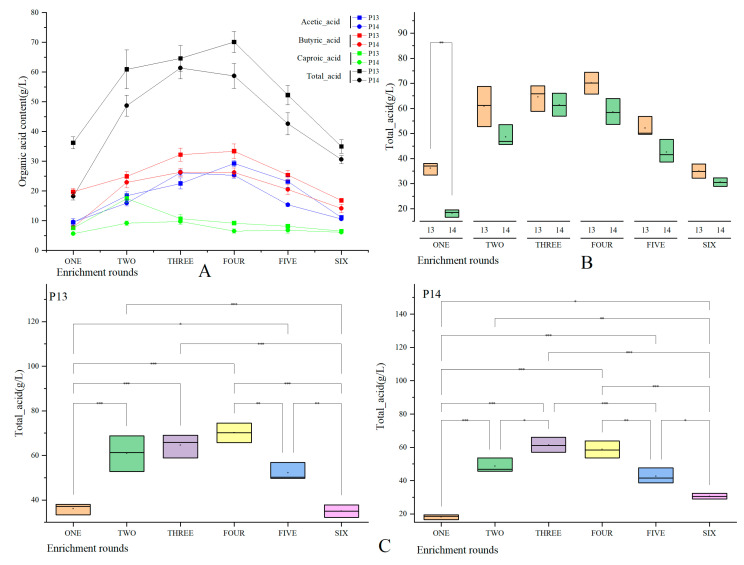
Organic acid content of enrichment-solution sample. (**A**) Changes in organic acids in P13 and P14 (total acids are weighted by volatile acid contents, including acetic acid, butyric acid, caproic acid, valeric acid, heptanoic acid). (**B**) Difference in total acids in EFB between heat-shock and non-heat-shock treatments in the 1st–6th rounds of enrichment period. (**C**) Difference in total acid accumulation between different enrichment cycles of P13 and P14 (more significant difference: * *p* ≤ 0.05; significant difference: ** *p* ≤ 0.01; extremely significant difference: *** *p* ≤ 0.001).

**Figure 2 foods-12-01594-f002:**
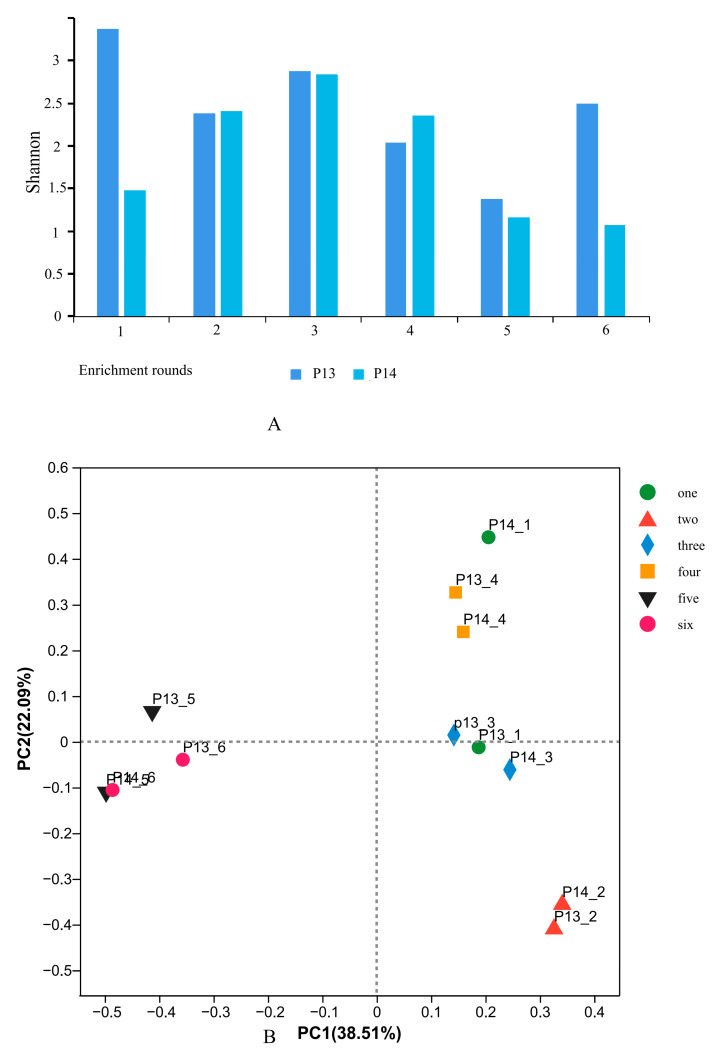
Microbial community diversities of EFB samples in different enrichment rounds. (**A**) Changes in Shannon diversity indexes of microorganisms in different enrichment cycles. (**B**) Principal coordinates analysis (PCoA) of microbial communities in different enrichment stages.

**Figure 3 foods-12-01594-f003:**
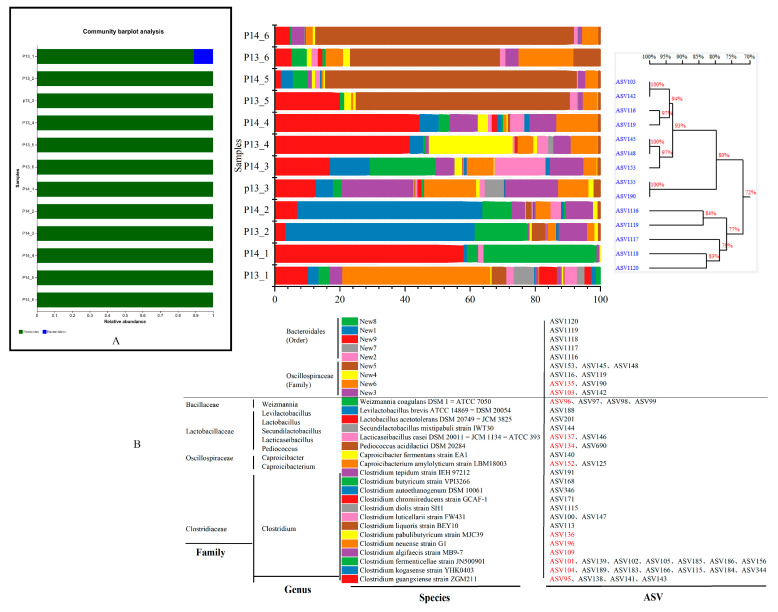
Distribution characteristics of bacterial community structures of samples in different enrichment rounds. (**A**) Percentage of community abundance at the phylum level. (**B**) Percentage of community abundance at the ASV level (the ASV was compared with the NCBI database, the species with a similarity of greater than 97% were grouped, and potential new species were classified and merged. Compared with the database, the similarity between ASV sequences is less than 97%, and the ASV sequences with similarity higher than 97% were classified as new species).

**Figure 4 foods-12-01594-f004:**
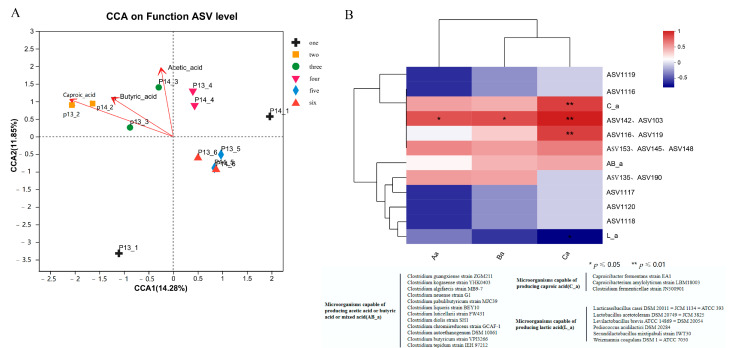
Relationship between microbial community and organic acid composition. (**A**) CCA analysis of functional microorganisms and organic acids. (**B**) Spearman analysis of functional microorganisms and metabolites (Aa: acetic acid; Ba: butyric acid; Ca: caproic acid; C_a: microorganisms capable of producing caproic acid; AB_a: microorganisms capable of producing acetic acid, butyric acid, or mixed acid; L_a: microorganisms capable of producing lactic acid). (more significant difference: * *p* ≤ 0.05; significant difference: ** *p* ≤ 0.01).

**Figure 5 foods-12-01594-f005:**
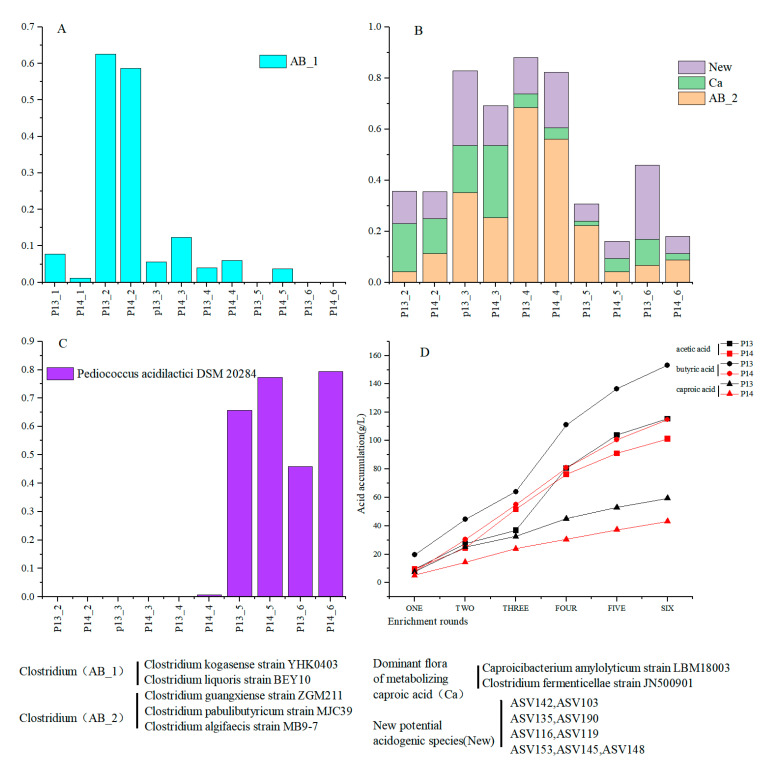
Changes in dominant microorganisms and acid accumulation in each period. (**A**) Changes in dominant acid-producing microorganisms in adaptation period. (**B**) Changes in dominant acid-producing microorganisms and potential new acid-producing species in main fermentation period. (**C**) The change rule of dominant acid-producing microorganisms in the later stage of fermentation (*Pediococcus acidilactici DSM 20284*). (**D**) Accumulation diagram of enriched acids in each round: the acid content in each round was derived from the accumulation of acid in that round plus the accumulation of acid in the previous round.

## Data Availability

The *16S rRNA* genome sequences of 12 samples were deposited in NCBI under the accession numbers SAMN32722911, SAMN32722912, SAMN32722913, SAMN32722914, SAMN32722915, SAMN32722916, SAMN32722917, SAMN32722918, SAMN32722919, SAMN32722920, SAMN32722921, and SAMN32722922, respectively. And other datasets generated during and analyzed during the current study are available from the corresponding author on reasonable request.

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
