# Peer review of "Effect of Multiple Rounds of Enrichment on Metabolite Accumulation and Microbiota Composition of Pit Mud for Baijiu Fermentation"

_foods, 2023, doi:10.3390/foods12081594_

Round 1

Reviewer 1 Report

This research article explored the effect of enrichment (six rounds) on the selection of functional microorganisms in Pit mud (PM) used for Chinese strong-flavor Baijiu preparation. The findings report the changes in metabolite accumulation and microbiota composition of PM.

It is a well-written article, with only minor correction needed. It would be better to enhance the figures' quality since it was difficult to read them (especially the text integrated). 

Author Response

Response to reviewers’ comments

Dear Editors and Reviewers,

Thank you for your letter and for the reviewers’ comments concerning our manuscript entitled “Effect of multiple rounds of enrichment on metabolite accumulation and microbiota composition of pit mud for Baijiu fermentation” (Manuscript ID: foods-2304843). Those comments are all valuable and very helpful for revising and improving our paper, as well as the important guiding significance to our researches. We have studied comments carefully and have made correction which we hope meet with approval. Revised portion are marked with different colors in the paper. The main corrections in the paper and the responds to the reviewer’s comments are as flowing:

Question 1: It is a well-written article, with only minor correction needed. It would be better to enhance the figures' quality since it was difficult to read them (especially the text integrated).

Answer: Thank you for your valuable and professional suggestion. We have replaced low-quality pictures with high-quality pictures.

Sincerely yours.

Reviewer 2 Report

Effect of multiple rounds of enrichment on metabolite accumulation and microbiota composition of pit mud for Baijiu fermentation

Fermentation products and microbial community composition of enriched fermented broth in different rounds of enrichment have been studied by the authors to determine the effect of the enrichment medium, enrichment rounds and heat shock treatment on the metabolites and microbial succession patterns in PM.

The manuscript is well-conceived and quite clear. Only few details are missing. Here are my comments:

1.       “the main the main” first line of the abstract, one of the “the main” should be omitted.

2.       The quality of the figures should be improved.

3.       Why the behavior of P13-1 and P14-1 (microbial communities in first enrichment stage) have shown significant difference compared to other enrichment stages (Fig. 2)?

Author Response

Response to reviewers’ comments

Dear Editors and Reviewers,

Thank you for your letter and for the reviewers’ comments concerning our manuscript entitled “Effect of multiple rounds of enrichment on metabolite accumulation and microbiota composition of pit mud for Baijiu fermentation” (Manuscript ID: foods-2304843). Those comments are all valuable and very helpful for revising and improving our paper, as well as the important guiding significance to our researches. We have studied comments carefully and have made correction which we hope meet with approval. Revised portion are marked with different colors in the paper. The main corrections in the paper and the responds to the reviewer’s comments are as flowing:

Question 1: “the main the main” first line of the abstract, one of the “the main” should be omitted.

Answer: Thank you for your valuable advice. We have deleted the first line of the abstract (the main the main). The font color at the modification is red.

Question 2: The quality of the figures should be improved.

Answer: Thank you for your comments. We have replaced all the pictures with high quality pictures.

Question 3: Why the behavior of P13-1 and P14-1 (microbial communities in first enrichment stage) have shown significant difference compared to other enrichment stages (Fig. 2)?

Answer: Thank you for your valuable questions. I have made a corresponding explanation in “3.2 Bacterial community diversity in PM-EFB” (The specific location is as follows: Lines 6-10, the background color of the text is yellow).

Sincerely yours.

Reviewer 3 Report

 Dear author,

your article is well done.

  • Is the manuscript clear, relevant for the field and presented in a well-structured manner?
    it is well structured. 
  • Are the cited references mostly recent publications (within the last 5 years) and relevant? Does it include an excessive number of self-citations?
    it is ok
  • Is the manuscript scientifically sound and is the experimental design appropriate to test the hypothesis? The scientific sound will be average and the design is appropiate
  • Are the manuscript’s results reproducible based on the details given in the methods section?
    technical yes but every pit mud
  • Are the figures/tables/images/schemes appropriate? Do they properly show the data? Are they easy to interpret and understand? Is the data interpreted appropriately and consistently throughout the manuscript? Please include details regarding the statistical analysis or data acquired from specific databases. The figures and tables are easy to understand
  • Are the conclusions consistent with the evidence and arguments presented?
    yes
  • Please evaluate the ethics statements and data availability statements to ensure they are adequate.

    I miss the ethical status - I think there will no problem because the topic  is about pit mud

  • Please explain pit mud ( because in translation you get soil suldge in Europe we use other terms) better in the indoduction and on page 13 speak about good wine.....do you really writing about wine or liquor?

    Best regards

Author Response

Dear Editors and Reviewers,

Thank you for your letter and for the reviewers’ comments concerning our manuscript entitled “Effect of multiple rounds of enrichment on metabolite accumulation and microbiota composition of pit mud for Baijiu fermentation” (Manuscript ID: foods-2304843). Those comments are all valuable and very helpful for revising and improving our paper, as well as the important guiding significance to our researches. We have studied comments carefully and have made correction which we hope meet with approval. Revised portion are marked with different colors in the paper. The main corrections in the paper and the responds to the reviewer’s comments are as flowing:

Question 1: Please explain pit mud (because in translation you get soil suldge in Europe we use other terms) better in the indoduction and on page 13 speak about good wine.....do you really writing about wine or liquor?

Answer: Glad to receive your valuable questions. We have made changes to good wine on page 13 (Specific as follows: Replace good wine with good SFB, SFB Means Chinese strong-flavor Baijiu, the font color at the modification is red).

Sincerely yours.
